# A Bayesian method for reducing bias in neural representational similarity analysis

**Ming Bo Cai**
Princeton Neuroscience Institute
Princeton University
Princeton, NJ 08544
`mcai@princeton.edu`

**Nicolas W. Schuck**
Princeton Neuroscience Institute
Princeton University
Princeton, NJ 08544
`nschuck@princeton.edu`

**Jonathan W. Pillow**
Princeton Neuroscience Institute
Princeton University
Princeton, NJ 08544
`pillow@princeton.edu`

**Yael Niv**
Princeton Neuroscience Institute
Princeton University
Princeton, NJ 08544
`yael@princeton.edu`

## Abstract

In neuroscience, the similarity matrix of neural activity patterns in response to different sensory stimuli or under different cognitive states reflects the structure of neural representational space. Existing methods derive point estimations of neural activity patterns from noisy neural imaging data, and the similarity is calculated from these point estimations. We show that this approach translates structured noise from estimated patterns into spurious bias structure in the resulting similarity matrix, which is especially severe when signal-to-noise ratio is low and experimental conditions cannot be fully randomized in a cognitive task. We propose an alternative Bayesian framework for computing representational similarity in which we treat the covariance structure of neural activity patterns as a hyper-parameter in a generative model of the neural data, and directly estimate this covariance structure from imaging data while marginalizing over the unknown activity patterns. Converting the estimated covariance structure into a correlation matrix offers a much less biased estimate of neural representational similarity. Our method can also simultaneously estimate a signal-to-noise map that informs where the learned representational structure is supported more strongly, and the learned covariance matrix can be used as a structured prior to constrain Bayesian estimation of neural activity patterns. Our code is freely available in Brain Imaging Analysis Kit (**Brainiak**) (https://github.com/IntelPNI/brainiak).

## 1   Neural pattern similarity as a way to understand neural representations

Understanding how patterns of neural activity relate to internal representations of the environment is one of the central themes of both system neuroscience and human neural imaging [20, 5, 7, 15]. One can record neural responses (e.g. by functional magnetic resonance imaging; fMRI) while participants observe sensory stimuli, and in parallel, build different computational models to mimic the brain's encoding of these stimuli. Neural activity pattern corresponding to each feature of an encoding model can then be estimated from the imaging data. Such activity patterns can be used to decode the perceived content with respect to the encoding features from new imaging data. The degree to which stimuli can be decoded from one brain area based on different encoding models informs us of the type of information represented in that area. For example, an encoding model based on motion energy in visual stimuli captured activity fluctuations from visual cortical areas V1 to V3,

and was used to successfully decode natural movie watched during an fMRI scan [14]. In contrast, encoding models based on semantic categories can more successfully decode information from higher level visual cortex [7].

While the decoding performance of different encoding models informs us of the type of information represented in a brain region, it does not directly reveal the structure of the representational space in that area. Such structure is indexed by how distinctively different contents are represented in that region [21, 4]. Therefore, one way to directly quantify the structure of the representational space in the neural population activity is to estimate the neural activity pattern elicited by each sensory stimulus, and calculate the similarity between the patterns corresponding to each pair of stimuli. This analysis of pair-wise similarity between neural activity patterns to different stimuli was named Representational Similarity Analysis (RSA) [11]. In fact, one of the earliest demonstrations of decoding from fMRI data was based on pattern similarity [7]. RSA revealed that the representational structures in the inferotemporal (IT) cortex of natural objects are highly similar between human and monkey [12] and a continuum in the abstract representation of biological classes exist in human ventral object visual cortex [2]. Because the similarity structure can be estimated from imaging data even without building an encoding model, RSA allows not only for model testing (by comparing the similarity matrix of neural data with the similarity matrix of the feature vectors when stimuli are represented with an encoding model) but also for exploratory study (e.g., by projecting the similarity structure to a low-dimensional space to visualize its structure, [11]). Therefore, originally as a tool for studying visual representations [2, 16, 10], RSA has recently attracted neuroscientists to explore the neural representational structure in many higher level cognitive areas [23, 18].

## 2   Structured noise in pattern estimation translates into bias in RSA

Although RSA is gaining popularity, a few recent studies revealed that in certain circumstances the similarity structure estimated by standard RSA might include a significant bias. For example, the estimated similarity between fMRI patterns of two stimuli is much higher when the stimuli are displayed closer in time [8]. This dependence of pattern similarity on inter-stimulus interval was hypothesized to reflect "temporal drift of pattern"[1], but we believe it may also be due to temporal autocorrelation in fMRI noise. Furthermore, we applied RSA to a dataset from a structured cognitive task (Fig **1A**) [19] and found that the highly structured representational similarity matrix obtained from the neural data (Fig **1B,C**) is very similar to the matrix obtained when RSA is applied to pure white noise (Fig **1D**). Since no task-related similarity structure should exist in white noise while the result in Fig **1D** is replicable from noise, this shows that the standard RSA approach can introduce similarity structure not present in the data.

We now provide an analytical derivation to explain the source of both types of bias (patterns closer in time are more similar and spurious similarity emerges from analyzing pure noise). It is notable that almost all applications of RSA explicitly or implicitly assume that fMRI responses are related to task-related events through a general linear model (GLM):

$$\mathbf{Y} = \mathbf{X} \cdot \boldsymbol{\beta} + \boldsymbol{\epsilon}. \tag{1}$$

Here, $\mathbf{Y} \in \mathbb{R}^{n_T \times n_S}$ is the fMRI time series from an experiment with $n_T$ time points from $n_S$ brain voxels. The experiment involves $n_C$ different conditions (e.g., different sensory stimuli, task states, or mental states), each of which comprises events whose onset time and duration is either controlled by the experimenter, or can be measured experimentally (e.g., reaction times). In fMRI, the measured blood oxygen-level dependent (BOLD) response is protracted, such that the response to condition $c$ is modelled as the time course of events in the experimental condition $s_c(t)$ convolved with a typical hemodynamic response function (HRF) $h(t)$. Importantly, each voxel can respond to different conditions with different amplitudes $\boldsymbol{\beta} \in \mathbb{R}^{n_C \times n_S}$, and the responses to all conditions are assumed to contribute linearly to the measured signal. Thus, denoting the matrix of HRF-convolved event time courses for each task condition with $\mathbf{X} \in \mathbb{R}^{n_T \times n_C}$, often called the *design matrix*, the measured $\mathbf{Y}$ is assumed to be a linear sum of $\mathbf{X}$ weighted by response amplitude $\boldsymbol{\beta}$ plus zero-mean noise.

Each row of $\boldsymbol{\beta}$ is the spatial response pattern (i.e., the response across voxels) to an experimental condition. The goal of RSA is therefore to estimate the similarity between the rows of $\boldsymbol{\beta}$. Because $\boldsymbol{\beta}$ is unknown, pattern similarity is usually calculated based on ordinary least square estimation of $\boldsymbol{\beta}$: $\hat{\boldsymbol{\beta}} = (\mathbf{X}^T\mathbf{X})^{-1}\mathbf{X}^T\mathbf{Y}$, and then using Pearson correlation of $\hat{\boldsymbol{\beta}}$ to measure similarity. Because

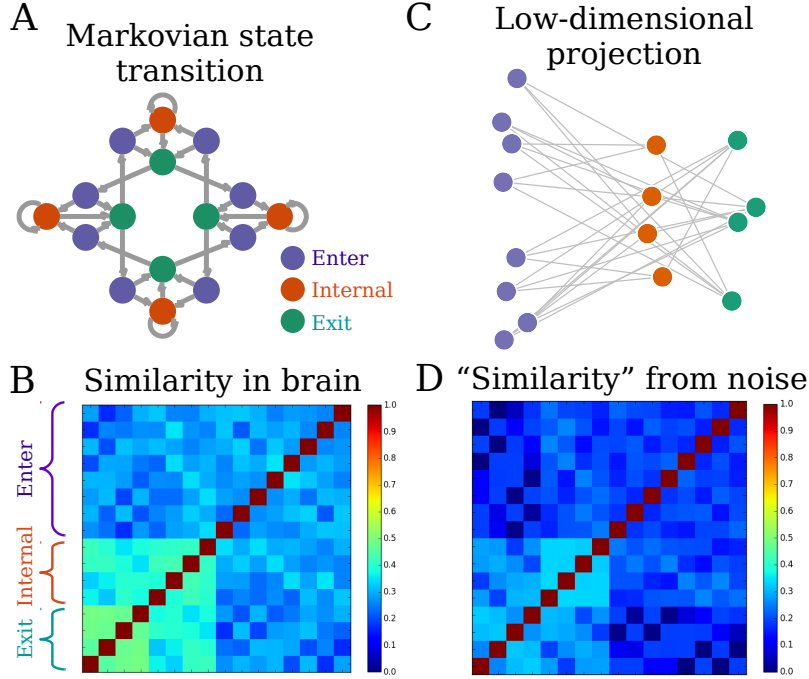

Figure 1: **Standard RSA introduces bias structure to the similarity matrix.** (A) A cognitive task that includes 16 different experimental conditions. Transitions between conditions follow a Markov process. Arrows indicate possible transitions, each with $p = 0.5$. The task conditions can be grouped to 3 categories (color coded) according to the semantics, or mental operations, required in each condition (the exact meaning of these conditions is not relevant to this paper). (B) Standard RSA of activity patterns corresponding to each condition estimated from a region of interest (ROI) reveal a highly structured similarity matrix. (C) Converting the similarity matrix $C$ to a distance matrix $1 - C$ and projecting it to a low-dimensional space using multi-dimensional scaling [13] reveals a highly regular structure. Seeing such a result, one may infer that representational structure in the ROI is strongly related to the semantic meanings of the task conditions. (D) However, a very similar similarity matrix can also be obtained if one applies standard RSA to pure white noise, with a similar low-dimensional projection (not shown). This indicates that standard RSA can introduce spurious structure in the resulting similarity matrix that does not exist in the data.

calculating sample correlation implies the belief that there exists an underlying covariance structure of $\beta$, we examine the source of bias by focusing on the covariance of $\hat{\beta}$ compared to that of true $\beta$.

We assume $\beta$ of all voxels in the ROI are indeed random vectors generated from a multivariate Gaussian distribution $\mathbf{N}(\mathbf{0}, \mathbf{U})$ (the size of $\mathbf{U}$ being $n_C \times n_C$). If one knew the true $\mathbf{U}$, similarity measures such as correlation could be derived from it. Substituting the expression $\mathbf{Y}$ from equation 1 we have $\hat{\beta} = \beta + (\mathbf{X}^T\mathbf{X})^{-1}\mathbf{X}^T\epsilon$. We assume that the signal $\beta$ is independent from the noise $\epsilon$, and therefore also independent from its linear transformation $(\mathbf{X}^T\mathbf{X})^{-1}\mathbf{X}^T\epsilon$. Thus the covariance of $\hat{\beta}$ is the sum of the true covariance of $\beta$ and the covariance of $(\mathbf{X}^T\mathbf{X})^{-1}\mathbf{X}^T\epsilon$:

$$\hat{\beta} \sim \mathbf{N}(0, U + (\mathbf{X}^T\mathbf{X})^{-1}\mathbf{X}^T\Sigma_\epsilon\mathbf{X}(\mathbf{X}^T\mathbf{X})^{-1}) \tag{2}$$

Where $\Sigma_\epsilon \in \mathbb{R}^{n_T \times n_T}$ is the temporal covariance of the noise $\epsilon$ (for illustration purposes, in this section we assume that all voxels have the same noise covariance).

The term

$$(\mathbf{X}^T\mathbf{X})^{-1}\mathbf{X}^T\Sigma_\epsilon\mathbf{X}(\mathbf{X}^T\mathbf{X})^{-1}$$

is the source of the bias. Since the covariance of $\hat{\beta}$ has this bias term adding to $\mathbf{U}$ which we are interested in, their sample correlation is also biased. So are many other similarity measures based on $\hat{\beta}$, such as Eucledian distance.

The bias term $(\mathbf{X}^T\mathbf{X})^{-1}\mathbf{X}^T\Sigma_\epsilon\mathbf{X}(\mathbf{X}^T\mathbf{X})^{-1}$ depends on both the design matrix and the properties of the noise. It is well known that autocorrelation exists in fMRI noise [24, 22]. Even if we assume that the noise is temporally independent (i.e., $\Sigma_\epsilon$ is a diagonal matrix, which may be a valid assumption if one "pre-whitens" the data before further analysis [22]), the bias structure still exists but reduces to $(\mathbf{X}^T\mathbf{X})^{-1}\sigma^2$, where $\sigma^2$ is the variance of the noise. Diedrichsen et al. [6] realized that the noise in $\hat{\boldsymbol{\beta}}$ could contribute to a bias in the correlation matrix but assumed the bias is only in the diagonal of the matrix. However, the bias is a diagonal matrix only if the columns of $\mathbf{X}$ (hypothetical fMRI response time courses to different conditions) are orthogonal to each other and if the noise has no autocorrelation. This is rarely the case for most cognitive tasks. In the example in Figure **1A**, the transitions between experimental conditions follow a Markov process such that some conditions are always temporally closer than others. Due to the long-lasting HRF, conditions of temporal proximity will have higher correlation in their corresponding columns in $\mathbf{X}$. Such correlation structure in $\mathbf{X}$ is the major determinant of the bias structure in this case. On the other hand, if each single stimulus is modelled as a condition in $\mathbf{X}$ and regularization is used during regression, the correlation between $\hat{\boldsymbol{\beta}}$ of temporally adjacent stimuli is higher primarily because of the autocorrelation property of the noise. This can be the major determinant of the bias structure in cases such as [8].

It is worth noting that the magnitude of bias is larger relative to the true covariance structure $\mathbf{U}$ when the signal-to-noise ratio (SNR) is lower, or when $\mathbf{X}$ has less power (i.e., there are few repetitions of each condition, thus few measurements of the related neural activity), as illustrated later in Figure **2B**.

The bias in RSA was not noticed until recently [1, 8], probably because RSA was initially applied to visual tasks in which stimuli are presented many times in a well randomized order. Such designs made the bias structure close to a diagonal matrix and researchers typically only focus on off-diagonal elements of a similarity matrix. In contrast, the neural signals in higher-level cognitive tasks are typically weaker than those in visual tasks [9]. Moreover, in many decision-making and memory studies the orders of different task conditions cannot be fully counter-balanced. Therefore, we expect the bias in RSA to be much stronger and highly structured in these cases, misleading researchers and hiding the true (but weaker) representational structure in the data.

One alternative to estimating $\hat{\boldsymbol{\beta}}$ using regression as above, is to perform RSA on the raw condition-averaged fMRI data (for instance, taking the average signal $\sim \mathbf{6}$ sec after the onset of an event as a proxy for $\hat{\boldsymbol{\beta}}$). This is equivalent to using a design matrix that assumes a 6-sec delayed single-pulse HRF. Although here columns of $\mathbf{X}$ are orthogonal by definition, the estimate $\hat{\boldsymbol{\beta}}$ is still biased, so is its covariance $(\mathbf{X}^T\mathbf{X})^{-1}\mathbf{X}^T\mathbf{X}_{true}U\mathbf{X}_{true}^T\mathbf{X}(\mathbf{X}^T\mathbf{X})^{-1} + (\mathbf{X}^T\mathbf{X})^{-1}\mathbf{X}^T\Sigma_\epsilon\mathbf{X}(\mathbf{X}^T\mathbf{X})^{-1}$ (where $\mathbf{X}_{true}$ is the design matrix reflecting the true HRF in fMRI). See supplementary material for illustration of this bias.

## 3 Maximum likelihood estimation of similarity structure directly from data

As shown in equation 2, the bias in RSA stems from treating the noisy estimate of $\boldsymbol{\beta}$ as the true $\boldsymbol{\beta}$ and performing a secondary analysis (correlation) on this noisy estimate. The similarly-structured noise (in terms of the covariance of their generating distribution) in each voxel's $\hat{\boldsymbol{\beta}}$ translates into bias in the secondary analysis. Since the bias comes from inferring $\mathbf{U}$ indirectly from point estimation of $\boldsymbol{\beta}$, a good way to avoid such bias is by not relying analysis on this point estimation. With a generative model relating $\mathbf{U}$ to the measured fMRI data $\mathbf{Y}$, we can avoid the point estimation of unknown $\boldsymbol{\beta}$ by marginalizing it in the likelihood of observing the data. In this section, we propose a method which performs maximum-likelihood estimation of the shared covariance structure $\mathbf{U}$ of activity patterns directly from the data.

Our generative model of fMRI data follows most of the assumptions above, but also allows the noise property and the SNR to vary across voxels. We use an AR(1) process to model the autocorrelation of noise in each voxel: for the $i^{th}$ voxel, we denote the noise at time $t(> 0)$ as $\epsilon_{t,i}$, and assume

$$\epsilon_{t,i} = \rho_i \cdot \epsilon_{t-1,i} + \eta_{t,i}, \quad \eta_{t,i} \sim \mathrm{N}(0, \sigma_i^2) \tag{3}$$

where $\sigma_i^2$ is the variance of the "new" noise and $\rho_i$ is the autoregressive coefficient for the $i^{th}$ voxel.

We assume that the covariance of the Gaussian distribution from which the activity amplitudes $\beta_i$ of the $i^{th}$ voxel are generated has a scaling factor that depends on its SNR $s_i$:

$$\beta_i \sim \mathrm{N}(0, (s_i\sigma_i)^2\mathbf{U}). \tag{4}$$

This is to reflect the fact that not all voxels in an ROI respond to tasks (voxels covering partially or entirely white matter might have little or no response). Because the magnitude of the BOLD response to a task is determined by the product of the magnitude of $X$ and $\beta$, but $s$ is a hyper-parameter only of $\beta$, we herefroth refer to $s$ as pseudo-SNR.

We further use the Cholesky decomposition to parametrize the shared covariance structure across voxels: $U = LL^T$, where $L$ is a lower triangular matrix. Thus, $\beta_i$ can be written as $\beta_i = s_i \sigma_i L \alpha_i$, where $\alpha_i \sim N(0, I)$ (this change of parameter allows for estimating $\mathbf{U}$ of less than full rank by setting $L$ as lower-triangular matrix with a few rightmost-columns truncated). And we have $Y_i - s_i \sigma_i X L \alpha_i \sim N(0, \Sigma_{\epsilon_i}(\sigma_i, \rho_i))$. Therefore, for the $i^{th}$ voxel, the likelihood of observing data $Y_i$ given the parameters is:

$$
\begin{aligned}
p(Y_i|L, \sigma_i, \rho_i, s_i) &= \int p(Y_i|L, \sigma_i, \rho_i, s_i, \alpha_i) p(\alpha_i) d\alpha_i \\
&= \int (2\pi)^{-\frac{n_T}{2}} |\Sigma_{\epsilon_i}^{-1}|^{\frac{1}{2}} exp[-\frac{1}{2}(Y_i - s_i \sigma_i X L \alpha_i)^T \Sigma_{\epsilon_i}^{-\frac{1}{2}}(Y_i - s_i \sigma_i X L \alpha_i)] \\
&\quad \cdot (2\pi)^{-\frac{n_C}{2}} exp[-\frac{1}{2}\alpha_i^T \alpha_i] d\alpha_i \\
&= (2\pi)^{-\frac{n_T}{2}} |\Sigma_{\epsilon_i}^{-1}|^{\frac{1}{2}} |\Lambda_i|^{\frac{1}{2}} exp[\frac{1}{2}((s_i \sigma_i)^2 Y_i^T \Sigma_{\epsilon_i}^{-1} X L \Lambda_i L^T X^T \Sigma_{\epsilon_i}^{-1} Y_i - Y_i^T \Sigma_{\epsilon_i}^{-1} Y_i)]
\end{aligned}
\tag{5}
$$

where $\Lambda_i = (s_i^2 \sigma_i^2 L^T X^T \Sigma_{\epsilon_i}^{-1} X L + I)^{-1}$. $\Sigma_{\epsilon_i}^{-1}$ is the inverse of the noise covariance matrix of the $i^{th}$ voxel, which is a function of $\sigma_i$ and $\rho_i$ (see supplementary material).

For simplicity, we assume that the noise for different voxels is independent, which is the common assumption of standard RSA (although see [21]). The likelihood of the whole dataset, including all voxels in an ROI, is then

$$
p(Y|L, \sigma, \rho, \mathbf{s}) = \prod_i p(Y_i|L, \sigma_i, \rho_i, s_i).
\tag{6}
$$

We can use gradient-based methods to optimize the model, that is, to search for the values of parameters that maximize the log likelihood of the data. Note that $\mathbf{s}$ are determined only up to a scale, because $L$ can be scaled down by a factor and all $s_i$ can be scaled up by the same factor without influencing the likelihood. Therefore, we set the geometric mean of $\mathbf{s}$ to be 1 to circumvent this indeterminacy, and fit $\mathbf{s}$ and $L$ iteratively. The spatial pattern of $\mathbf{s}$ thus only reflects the relative SNR of different voxels.

Once we obtain $\hat{L}$, the estimate of $L$, we can convert the covariance matrix $\hat{U} = \hat{L}\hat{L}^T$ into a correlation matrix, which is our estimation of neural representational similarity. Because $U$ is a hyper-parameter of the activity pattern in our generative model and we estimate it directly from data, this is an empirical Bayesian approach. We therefore refer to our method as "Bayesian RSA" now.

## 4 Performance of the method

### 4.1 Reduced bias in recovering the latent covariance structure from simulated data

To test if the proposed method indeed reduces bias, we simulated fMRI data with a predefined covariance structure and compared the structure recovered by our method with that recovered by standard RSA. Fig **2A** shows the hypothetical covariance structure from which we drew $\beta_i$ for each voxel. The bias structure in Fig **1D** is the average structure induced by the design matrices of all participants. To simplify the comparison, we use the design matrices of the experiment experienced by one participant. As a result, the bias structure induced by the design matrix deviates slightly from that in Fig **1D**.

As mentioned, the contribution of the bias to the covariance of $\hat{\beta}$ depends on both the level of noise and the power in the design matrix $X$. The more each experimental condition is measured during an experiment (roughly speaking, the longer the experiment), the less noisy the estimation of $\hat{\beta}$, and the less biased the standard RSA is. To evaluate the improvement of our method over standard RSA

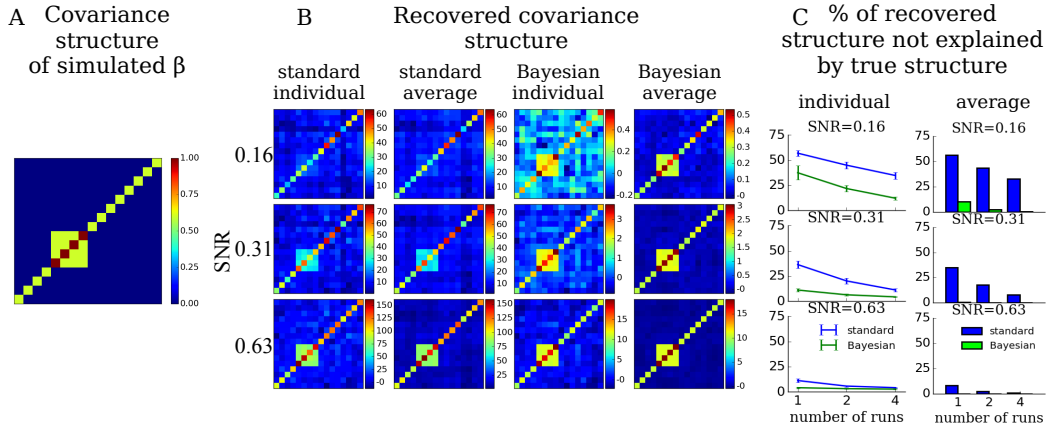

Figure 2: **Bayesian RSA reduces bias in the recovered shared covariance structure of activity patterns.** (A) The covariance structure from which we sampled neural activity amplitudes $\beta$ for each voxel. fMRI data were synthesized by weighting the design matrix of the task from Fig **1A** with the simulated $\beta$ and adding AR(1) noise. (B) The recovered covariance structure for different simulated pseudo-SNR. Standard individual: covariance calculated directly from $\hat{\beta}$ as is done in standard RSA, for one simulated participant. Standard average: average of covariance matrices of $\hat{\beta}$ from 20 simulated participants. Bayesian individual: covariance estimated directly from data by our method for one simulated participant. Bayesian average: average of the covariance matrices estimated by Bayesian RSA from 20 simulated participants. (C) The ratio of the variation in the recovered covariance structure which cannot be explained by the true covariance structure in Fig **2A**. Left: the ratio for covariance matrix from individual simulation (panel 1 and 3 of Fig **2B**). Right: the ratio for average covariance matrix (panel 2 and 4 of Fig **2B**). Number of runs: the design matrices of 1, 2, or 4 runs of a participant in the experiment of Fig **1A** were used in each simulation, to test the effect of experiment duration. Error bar: standard deviation.

in different scenarios, we therefore varied two factors: the average SNR of voxels and the duration of the experiment. 500 voxels were simulated. For each voxel, $\sigma_i$ was sampled uniformly from $[1.0, 3.0]$, $\rho_i$ was sampled uniformly from $[-0.2, 0.6]$ (our empirical investigation of example fMRI data shows that small negative autoregressive coefficient can occur in white matter), $s_i$ was sampled uniformly from $f \cdot [0.5, 2.0]$. The average SNR was manipulated by choosing $f$ from one of three levels $\{1, 2, 4\}$ in different simulations. The duration of the experiment was manipulated by using the design matrices of run 1, runs 1-2, and runs 1-4 from one participant.

Fig **2B** displays the covariance matrix recovered by standard RSA (first two columns) and Bayesian RSA (last two columns), with an experiment duration of approximately 10 minutes (one run, measurement resolution: TR = 2.4 sec). The rows correspond to different levels of average SNR (calculated post-hoc by averaging the ratio $\frac{std(X\beta_i)}{\sigma_i}$ across voxels). Covariance matrices recovered from one simulated participant and the average of covariance matrices recovered from 20 simulated participants ("average") are displayed. Comparing the shapes of the matrix and the magnitudes of values (color bars) across rows, one can see that the bias structure in standard RSA is most severe when SNR is low. Averaging the estimated covariance matrices across simulated participants can reduce noise, but not bias. Comparing between columns, one can see that strong residual structure exists in standard RSA even after averaging, but almost disappears for Bayesian RSA. This is especially apparent for low SNR – the block structure of the true covariance matrix from Figure **2A** is almost undetectable for standard RSA even after averaging (column 2, row 1 of Fig **2B**), but emerges after averaging for Bayesian RSA (column 4, row 1 of Fig **2B**). Fig **2C** compares the proportion of variation in the recovered covariance structure that cannot be explained by the true structure in Fig **2A**, for different levels of SNR and different experiment durations, for individual simulated participants and for average results. This comparison confirms that the covariance recovered by Bayesian RSA deviates much less from the true covariance matrix than that by standard RSA, and that the deviation observed in an individual participant can be reduced considerably by averaging over multiple participants (comparing the left with right panels of Fig **2C** for Bayesian RSA).

## 4.2 Application to real data: simultaneous estimation of neural representational similarity and spatial location supporting the representation

In addition to reducing bias in estimation of representational similarity, our method also has an advantage over standard RSA: it estimates the pseudo-SNR map **s**. This map reveals the locations within the ROI that support the identified representational structure. When a researcher looks into an anatomically defined ROI, it is often the case that only some of the voxels respond to the task conditions. In standard RSA, $\hat{\beta}$ in voxels with little or no response to tasks is dominated by structured noise following the bias covariance structure $(\mathbf{X}^T\mathbf{X})^{-1}\mathbf{X}^T\Sigma_\epsilon\mathbf{X}(\mathbf{X}^T\mathbf{X})^{-1}$, but all voxels are taken into account equally in the analysis. In contrast, $s_i$ in our model is a hyper-parameter learned directly from data – if a voxel does not respond to any condition of the task, $s_i$ would be small and the contribution of the voxel to the total log likelihood is small. The fitting of the shared covariance structure is thus less influenced by this voxel.

From our simulated data, we found that parameters of the noise ($\sigma$ and $\rho$) can be recovered reliably with small variance. However, the estimation of **s** had large variance from the true values used in the simulation. One approach to reduce variance of estimation is by harnessing prior knowledge about data. Voxels supporting similar representation of sensory input or tasks tend to spatially cluster together. Therefore, we used a Gaussian Process to impose a smooth prior on $log(s)$ [17]. Specifically, for any two voxels i and j, we assumed $cov(log(s_i), log(s_j)) = b^2 exp(-\frac{(x_i-x_j)^T(x_i-x_j)}{2l_{space}^2} - \frac{(I_i-I_j)^2}{2l_{inten}^2})$, where $x_i$ and $x_j$ are the spatial coordinates of the two voxels and $I_i$ and $I_j$ are the average intensities of fMRI signals of the two voxels. Intuitively, this means that if two voxels are close together and have similar signal intensity (that is, they are of the same tissue type), then they should have similar SNR. Such a kernel of a Gaussian Process imposes spatial smoothness but also allows the pseudo-SNR to change quickly at tissue boundaries. The variance of the Gaussian process $b^2$, the length scale $l_{space}$ and $l_{inten}$ were fitted together with the other parameters by maximizing the joint log likelihood of all parameters (here again, we restrict the geometric mean of **s** to be 1).

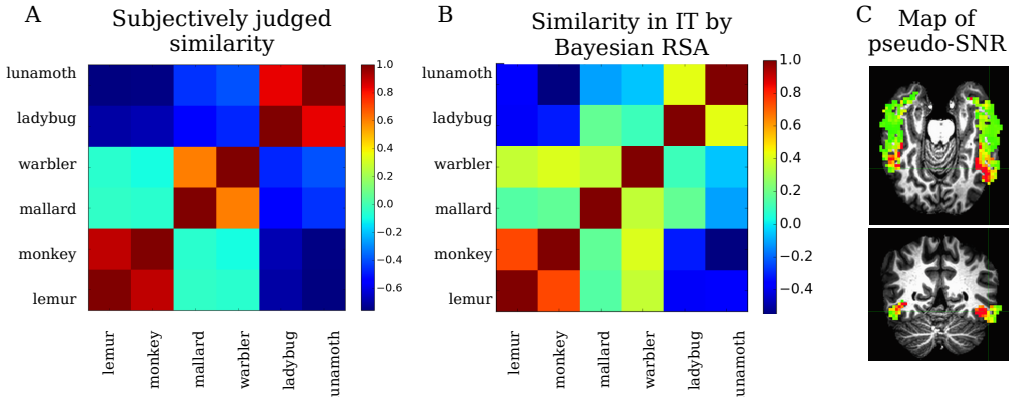

Figure 3: **Bayesian RSA estimates both the representational similarity structure from fMRI data and the spatial map supporting the learned representation.** (A) Similarity between 6 animal categories, as judged behaviorally (reproduced from [2]). (B) Average representational similarity estimated from IT cortex from all participants of [2], using our approach. The estimated structure resembles the subjectively-reported structure. (C) Pseudo-SNR map in IT cortex corresponding to one participant. Red: high pseudo-SNR, green: low pseudo-SNR. Only small clusters of voxels show high pseudo-SNR.

We applied our method to the dataset of Connolly et al. (2012) [2]. In their experiment, participants viewed images of animals from 6 different categories during an fMRI scan and rated the similarity between animals outside the scanner. fMRI time series were pre-processed in the same way as in their work [2]. Inferior temporal (IT) cortex is generally considered as the late stage of ventral pathway of the visual system, in which object identity is represented. Fig **3** shows the similarity judged by the participants and the average similarity matrix estimated from IT cortex, which shows similar structure but higher correlations between animal classes. Interestingly, the pseudo-SNR map shows that only part of the anatomically-defined ROI supports the representational structure.

# 5    Discussion

In this paper, we demonstrated that representational similarity analysis, a popular method in many recent fMRI studies, suffers from a bias. We showed analytically that such bias is contributed by both the structure of the experiment design and the covariance structure of measurement and neural noise. The bias is induced because standard RSA analyzes noisy estimates of neural activation level, and the structured noise in the estimates turns into bias. Such bias is especially severe when SNR is low and when the order of task conditions cannot be fully counterbalanced. To overcome this bias, we proposed a Bayesian framework of the fMRI data, incorporating the representational structure as the shared covariance structure of activity levels across voxels. Our Bayesian RSA method estimates this covariance structure directly from data, avoiding the structured noise in point estimation of activity levels. Our method can be applied to neural recordings from other modalities as well.

Using simulated data, we showed that, as compared to standard RSA, the covariance structure estimated by our method deviates much less from the true covariance structure, especially for low SNR and short experiments. Furthermore, our method has the advantage of taking into account the variation in SNR across voxels. In future work, we will use the pseudo-SNR map and the covariance structure learned from the data jointly as an empirical prior to constrain the estimation of activation levels $\beta$. We believe that such structured priors learned directly from data can potentially provide more accurate estimation of neural activation patterns—the bread and butter of fMRI analyses.

A number of approaches have recently been proposed to deal with the bias structure in RSA, such as using the correlation or Mahalanobis distance between neural activity patterns estimated from separate fMRI scans instead of from the same fMRI scan, or modeling the bias structure as a diagonal matrix or by a Taylor expansion of an unknown function of inter-events intervals [1, 21, 6]. Such approaches have different limitations. The correlation between patterns estimated from different scans [1] is severely underestimated if SNR is low (for example, unless there is zero noise, the correlation between the neural patterns corresponding to the same conditions estimated from different fMRI scans is always smaller than 1, while the true patterns should presumably be the same across scans in order for such an analysis to be justified). Similar problems exists for using Mahalanobis distance between patterns estimated from different scans [21]: with noise in the data, it is not guaranteed that the distance between patterns of the same condition estimated from separate scans is smaller than the distance between patterns of different conditions. Such a result cannot be interpreted as a measure of "similarity" because, theoretically, neural patterns should be more similar if they belong to the same condition than if they belong to different conditions. Our approach does not suffer from such limitations, because we are directly estimating a covariance structure, which can always be converted to a correlation matrix. Modeling the bias as a diagonal matrix [6] is not sufficient, as the bias can be far from diagonal, as shown in Fig **1D**. Taylor expansion of the bias covariance structure as a function of inter-event intervals can potentially account for off-diagonal elements of the bias structure, but it has the risk of removing structure in the true covariance matrix if it happens to co-vary with inter-event intervals, and becomes complicated to set up if conditions repeat multiple times [1].

One limitation of our model is the assumption that noise is spatially independent. Henriksson et al. [8] suggested that global fluctuations of fMRI time series over large areas (which is reflected as spatial correlation) might contribute largely to their RSA pattern. This might also be the reason that the overall correlation in Fig **1B** is higher than the bias obtained from standard RSA on independent Gaussian noise (Fig **1D**). Our future work will explicitly incorporate such global fluctuations of noise.

## Acknowledgement

This publication was made possible through the support of grants from the John Templeton Foundation and the Intel Corporation. The opinions expressed in this publication are those of the authors and do not necessarily reflect the views of the John Templeton Foundation. JWP was supported by grants from the McKnight Foundation, Simons Collaboration on the Global Brain (SCGB AWD1004351) and the NSF CAREER Award (IIS-1150186). We thank Andrew C. Connolly etc. for sharing of the data used in **4.2**. Data used in the supplementary material were obtained from the MGH-USC Human Connectome Project (HCP) database.

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
