[Supplementary Material]

# Supplementary material

## S.1 Parametrization of noise covariance

We denote the noise of voxel $i$ at time point $t$ by $\varepsilon_{i,t}$, and denote the temporal covariance of the noise time series $\varepsilon_i$ as $\Sigma_{\epsilon_i}$ .

Following AR(1) model,

$$\epsilon_{i,t}=\rho_i\,\epsilon_{i,t-1}+\eta_{i,t} \tag{s-1}$$

where $\eta_{i,t}\sim N(0,\sigma_i^2)$ is the "new" noise.

Then we have $\epsilon_t-\rho_i\,\epsilon_{t-1}=\eta_t$ .

Denote $B_i=I-\rho_i P_1$ , where $P_1=\begin{bmatrix} 0 & 0 & 0 & & \\ 1 & 0 & 0 & & \cdots \\ 0 & 1 & 0 & & \\ & & & 0 & 0 \\ & \cdots & & 1 & 0 \end{bmatrix}$ . Multiplying matrix $B_i$ to the noise $\epsilon_i$

removes the temporal correlation: $B_i\,\epsilon_i\sim N(0,\sigma_i^2\hat{I})$ .

$\hat{I}_i$ is the same as an identity matrix except that its (1,1) element is $\dfrac{1}{1-\rho_i^2}$ , because the variance of

the AR(1) noise at any time point without knowing the noise at previous time points is $\dfrac{\sigma_i^2}{1-\rho_i^2}$ .

We also know that $B_i\,\epsilon_i\sim N(0,B_i\Sigma_{\epsilon_i}B_i^T)$

Therefore, $B_i\Sigma_{\epsilon_i}B_i^T=\sigma_i^2\hat{I}_i$ , $\Sigma_Y=\sigma^2 B_i^{-1}\hat{I}_i(B_i^T)^{-1}$

and

$$\Sigma_Y^{-1}=\frac{1}{\sigma^2}B_i^T\hat{I}_i^{-1}B_i \tag{s-2}$$

$\hat{I}_i^{-1}$ has $1-\rho_i^2$ on element (1,1). In other words, $\hat{I}_i^{-1}=I-\rho_i^2\delta_1\delta_1^T$ . Here we use $\delta_k$ to denote a unit vector which is 1 at the k-th element and 0 in other elements.

We expand (s-2):

$$\Sigma_{\epsilon_i}^{-1}=\sigma_i^{-2}B_i^T\hat{I}_i^{-1}B_i$$
$$=\sigma_i^{-2}(I-\rho_i P_1)^T(I-\rho_i^2\delta_1\delta_1^T)(I-\rho_i P_1)$$
$$=\sigma_i^{-2}[I-\rho_i P_1^T-\rho_i^2\delta_1\delta_1^T+\rho_i^3 P_1^T\delta_1\delta_1^T-\rho_i P_1+\rho_i^2 P_1^T P_1+\rho_i^3\delta_1\delta_1^T P_1-\rho_i^4 P_1^T\delta_1\delta_1^T P_1]$$
$$=\sigma_i^{-2}[I-\rho_i(P_1^T+P_1)+\rho_i^2(I-\delta_1\delta_1^T-\delta_n\delta_n^T)]$$

The last equality is because $P_1^T\delta_1$ is a zero vector, and that $P_1^T P_1=I-\delta_n\delta_n^T$

We can write $D=P_1^T+P_1$ and $F=I-\delta_1\delta_1^T-\delta_n\delta_n^T$

Then $D$ and $F$ are two template matrices which do not depend on any free parameters.

So we can write the inverse covariance matrix as

$$\Sigma_{\epsilon_i}^{-1}=\sigma_i^{-2}[I-\rho_i D+\rho_i^2 F] \tag{s-3}$$

We can further denote $A_i=A(\rho_i)=I-\rho_i D+\rho_i^2 F$ , so $\Sigma_{\epsilon_i}^{-1}=\sigma_i^{-2}A_i$ (s-4)

Because $|B_i|=1$ and $|\hat{I}_i^{-1}|=1-\rho_i^2$ , we have $|\Sigma_{\epsilon_i}^{-1}|=\sigma_i^{-2n_T}|B_i||\hat{I}_i^{-1}||B_i|=\sigma_i^{-2n_T}(1-\rho_i^2)$

## S.2 Derivation of log likelihood

Following the notation in the article, for voxel $i$, we reparametrize $\beta_i=(s_i\sigma_i)L\alpha_i, \alpha_i\sim N(0,I_r)$ , where $r$ is the rank of $U=LL^T$ , the shared covariance matrix. The likelihood of observing data $Y_i$ (time series of voxel $i$) is:

$$p(Y_i|L,\alpha_i,s_i\sigma_i,\rho_i)p(\alpha_i)$$

$$=N(Y_i;s_i\sigma_i XL\alpha_i,\rho_i)N(\alpha_i;0,I_r)$$

$$=(2\pi)^{-\frac{n_T}{2}}|\Sigma_{\epsilon_i}|^{-\frac{1}{2}}\exp[-\frac{1}{2}(Y_i-s_i\sigma_i XL\alpha_i)^T\Sigma_{\epsilon_i}^{-1}(Y_i-s_i\sigma_i XL\alpha_i)](2\pi)^{-\frac{r}{2}}\exp[-\frac{1}{2}\alpha_i^T\alpha_i]$$

$$=(2\pi)^{-\frac{(n_T+r)}{2}}|\Sigma_{\epsilon_i}|^{-\frac{1}{2}}\exp[-\frac{1}{2}(s_i^2\sigma_i^2\alpha_i^T L^T X^T \Sigma_{\epsilon_i}^{-1}XL\alpha_i-2s_i\sigma_i Y_i^T\Sigma_{\epsilon_i}^{-1}XL\alpha_i+Y_i^T\Sigma_{\epsilon_i}^{-1}Y_i+\alpha_i^T\alpha_i)]$$

$$=(2\pi)^{-\frac{(n_T+r)}{2}}|\Sigma_{\epsilon_i}|^{-\frac{1}{2}}\exp[-\frac{1}{2}(\alpha_i^T(s_i^2\sigma_i^2 L^T X^T \Sigma_{\epsilon_i}^{-1}XL+I)\alpha_i-2s_i\sigma_i Y_i^T\Sigma_{\epsilon_i}^{-1}XL\alpha_i+Y_i^T\Sigma_{\epsilon_i}^{-1}Y_i)]$$

$$=(2\pi)^{-\frac{(n_T+r)}{2}}|\Sigma_{\epsilon_i}|^{-\frac{1}{2}}\exp\{-\frac{1}{2}[\alpha_i^T\Lambda_i^{-1}\alpha_i-2(s_i\sigma_i\Lambda_i L^T X^T \Sigma_{\epsilon_i}^{-1}Y_i)^T\Lambda_i^{-1}\alpha_i+Y_i^T\Sigma_{\epsilon_i}^{-1}Y_i]\}$$

$$=(2\pi)^{-\frac{(n_T+r)}{2}}|\Sigma_{\epsilon_i}|^{-\frac{1}{2}}\exp[-\frac{1}{2}(\alpha_i-\mu_i)^T\Lambda_i^{-1}(\alpha_i-\mu_i)+\frac{1}{2}(s_i^2\sigma_i^2 Y_i^T\Sigma_{\epsilon_i}^{-1}XL\Lambda_i L^T X^T \Sigma_{\epsilon_i}^{-1}Y_i-Y_i^T\Sigma_{\epsilon_i}^{-1}Y_i)]$$

$$=(2\pi)^{-\frac{n_T}{2}}|\Sigma_{\epsilon_i}|^{-\frac{1}{2}}|\Lambda_i|^{\frac{1}{2}}\exp[\frac{1}{2}(s_i^2\sigma_i^2 Y_i^T\Sigma_{\epsilon_i}^{-1}XL\Lambda_i L^T X^T \Sigma_{\epsilon_i}^{-1}Y_i-Y_i^T\Sigma_{\epsilon_i}^{-1}Y_i)]N(\alpha_i;\mu_i,\Lambda_i)$$

$$=(2\pi)^{-\frac{n_T}{2}}|\Sigma_{\epsilon_i}|^{-\frac{1}{2}}|\Lambda_i|^{\frac{1}{2}}\exp[\frac{1}{2}(\mu_i^T\Lambda_i^{-1}\mu_i-Y_i^T\Sigma_{\epsilon_i}^{-1}Y_i)]N(\alpha_i;\mu_i,\Lambda_i)$$

**(s-5)**

In the derivation above,
$\mu_i=s_i\sigma_i\Lambda_i L^T X^T \Sigma_{\epsilon_i}^{-1}Y_i$ and $\Lambda_i=(s_i^2\sigma_i^2 L^T X^T \Sigma_{\epsilon_i}^{-1}XL+I)^{-1}$ are the posterior mean and variance of $\alpha_i$ .

$I$ is identity matrix of size $r$ by $r$, where r is the rank of $U$, and the number of columns of $L$.
By plugging in (s-4), we get
$$\Lambda_i=(s_i^2 L^T X^T A_i XL+I)^{-1}$$

$$\mu_i=\frac{s_i}{\sigma_i}\Lambda_i L^T X^T A_i Y_i$$

**(s-6)**

Marginalizing $\alpha_i$ :
$$p(Y_i|L,s_i\sigma_i,\rho_i)$$
$$=\int d\alpha_i p(Y_i|L,\alpha_i,s_i\sigma_i,\rho_i)p(\alpha_i)$$

$$=\int d\alpha_i N(\alpha_i;\mu_i,\Lambda_i)(2\pi)^{-\frac{n_T}{2}}|\Sigma_{\epsilon_i}|^{-\frac{1}{2}}|\Lambda_i|^{\frac{1}{2}}\exp[\frac{1}{2}(\mu_i^T\Lambda_i^{-1}\mu_i-Y_i^T\Sigma_{\epsilon_i}^{-1}Y_i)]$$

$$=(2\pi)^{-\frac{n_T}{2}}|\Sigma_{\epsilon_i}|^{-\frac{1}{2}}|\Lambda_i|^{\frac{1}{2}}\exp[\frac{1}{2}(\mu_i^T\Lambda_i^{-1}\mu_i-Y_i^T\Sigma_{\epsilon_i}^{-1}Y_i)]$$

Therefore, the marginal log likelihood is

$$\log\left(p\left(Y_i|L, s_i\, \sigma_i, \rho_i\right)\right)$$

$$= \log\left\{ \left(2\pi\right)^{-\frac{n_T}{2}} |\Sigma_{\epsilon_i}|^{-\frac{1}{2}} |\Lambda_i|^{\frac{1}{2}} \exp\left[\frac{1}{2}\left(\mu_i^T \Lambda_i^{-1} \mu_i - Y_i^T \Sigma_{\epsilon_i}^{-1} Y_i\right)\right]\right\}$$

$$= -\frac{n_T}{2}\log\left(2\pi\right) - \frac{n_T}{2}\log\sigma_i^2 + \frac{1}{2}\log\left(1-\rho_i^2\right) - \frac{1}{2}\log\left|s_i^2 L^T X^T A_i X L + I\right| \qquad \text{(s-7)}$$

$$+ \frac{1}{2\sigma_i^2}\left[s_i^2 Y_i^T A_i X L \Lambda_i L^T X^T A_i Y_i - Y_i^T A_i Y_i\right]$$

If there are multiple runs, the temporal covariance matrix becomes block diagonal matrix, with each block on the diagonal corresponding to the covariance matrix of one run. Its determinant is the product of the determinant of each block on the diagonal. Therefore, the term $\frac{1}{2}\log\left(1-\rho_i^2\right)$ becomes

$\frac{n_{run}}{2}\log\left(1-\rho_i^2\right)$ , where $n_{run}$ is the number of fMRI runs.

$$\log\left(p\left(Y_i|L, s_i\, \sigma_i, \rho_i\right)\right)$$

$$= -\frac{n_T}{2}\log\left(2\pi\right) - \frac{n_T}{2}\log\sigma_i^2 + \frac{n_{run}}{2}\log\left(1-\rho_i^2\right) - \frac{1}{2}\log\left|\Lambda_i^{-1}\right| \qquad \text{(s-8)}$$

$$+ \frac{1}{2\sigma_i^2}\left[s_i^2 Y_i^T A_i X L \Lambda_i L^T X^T A_i Y_i - Y_i^T A_i Y_i\right]$$

In our implementation, we parametrize $s_i$ with $\log(s_i^2)$, $\sigma_i$ with $\log(\sigma_i^2)$, and $\rho_i$ with $a_i = \tan\left(\frac{\pi}{2}\rho_i\right)$ . In this way, all free parameters become unbounded and we can use unrestricted gradient based algorithm to maximize the marginal log likelihood.

# S.3 Gradient of the log likelihood

### S.3.1 Gradient with respect to $\sigma_i^2$

$$\frac{\partial\log\left(p\left(Y_i|L, s_i\, \sigma_i, \rho_i\right)\right)}{\sigma_i^2}$$

$$= -\frac{n_T}{2\sigma_i^2} - \frac{1}{2\sigma_i^4}\left[s_i^2 Y_i^T A_i X L \Lambda_i L^T X^T A_i Y_i - Y_i^T A_i Y_i\right]$$

The maximum likelihood estimate of $\sigma_i^2$ given other parameters is achieved when the gradient above equals zero. Therefore,

$$\sigma_{i,ML}^2 = \frac{Y_i^T A_i Y_i - s_i^2 Y_i^T A_i X L \Lambda_i L^T X^T A_i Y_i}{n_T} \qquad \text{(s-9)}$$

So we do not need to pursue gradient descent on $\sigma_i^2$. This reduces the number of free parameters by close to one third.

## S.3.2 Gradient with respect to log($s_i^2$)

$$\frac{\partial \log\left(p\left(Y_i | L, s_i\, \sigma_i, \rho_i\right)\right)}{\partial s_i^2}$$

$$=-\frac{1}{2}Tr\left[\left(s_i^2 L^T X^T A_i X L + I\right)^{-1}\frac{\partial\left(s_i^2 L^T X^T A_i X L + I\right)}{\partial s_i^2}\right]$$

$$+\frac{1}{2}\sigma_i^{-2}Y_i^T A_i X L\left[s_i^2 L^T X^T A_i X L + I\right]^{-1}L^T X^T A_i Y_i$$

$$-\frac{1}{2}\sigma_i^{-2}s_i^2 Y_i^T A_i X L\left[s_i^2 L^T X^T A_i X L + I\right]^{-1}$$

$$\cdot\frac{\partial\left(s_i^2 L^T X^T A_i X L + I\right)}{\partial s_i^2}\cdot\left[s_i^2 L^T X^T A_i X L + I\right]^{-1}L^T X^T A_i Y_i$$

$$=-\frac{1}{2}Tr\left[\left(s_i^2 L^T X^T A_i X L + I\right)^{-1}L^T X^T A_i X L\right]$$

$$+\frac{1}{2}\sigma_i^{-2}Y_i^T A_i X L \Lambda_i L^T X^T A_i Y_i$$

$$-\frac{1}{2}\sigma_i^{-2}s_i^2 Y_i^T A_i X L \Lambda_i L^T X^T A_i X L \Lambda_i L^T X^T A_i Y_i$$

$$=-\frac{1}{2}Tr\left[\left(s_i^2 L^T X^T A_i X L + I\right)^{-1}\left(s_i^2 L^T X^T A_i X L + I - I\right)s_i^{-2}\right]$$

$$+\frac{1}{2}\sigma_i^{-2}Y_i^T A_i X L\left(\Lambda_i - s_i^2 \Lambda_i L^T X^T A_i X L \Lambda_i\right)L^T X^T A_i Y_i$$

$$=-\frac{1}{2s_i^2}Tr\left[I - \left(s_i^2 L^T X^T A_i X L + I\right)^{-1}\right]$$

$$+\frac{1}{2}\sigma_i^{-2}Y_i^T A_i X L\left(\Lambda_i - \Lambda_i\left(I + s_i^2 L^T X^T A_i X L - I\right)\Lambda_i\right)L^T X^T A_i Y_i$$

$$=\frac{1}{2s_i^2}\left(-r + Tr[\Lambda_i]\right)+\frac{1}{2}\sigma_i^{-2}Y_i^T A_i X L\left(\Lambda_i - \Lambda_i\left(\Lambda_i^{-1} - I\right)\Lambda_i\right)L^T X^T A_i Y_i$$

$$=\frac{1}{2s_i^2}\left(-r + Tr[\Lambda_i]\right)+\frac{1}{2}\sigma_i^{-2}Y_i^T A_i X L \Lambda_i^2 L^T X^T A_i Y_i$$

Therefore,

$$\frac{\partial \log\left(p\left(Y_i | L, s_i\, \sigma_i, \rho_i\right)\right)}{\partial \log s_i^2}$$

$$=\frac{\partial \log\left(p\left(Y_i | L, \sigma_i, \rho_i, s_i\right)\right)}{\partial s_i^2}\frac{\partial s_i^2}{\partial \log s_i^2} \qquad\qquad \textbf{(s-10)}$$

$$=\frac{1}{2}\left(-r + Tr[\Lambda_i]\right)+\frac{s_i^2}{2\sigma_i^2}Y_i^T A_i X L \Lambda_i^2 L^T X^T A_i Y_i$$

## S.3.3 Gradient with respect to $\quad a_i = \tan\left(\frac{\pi}{2}\rho_i\right)$

$$\frac{\partial a_i}{\partial \rho_i} = \frac{2}{\pi(1+a_i^2)}$$

The derivative of $A_i$ with respect to $\rho_i$ is:

$$\frac{\partial A_i}{\partial \rho_i} = \frac{\partial (I - \rho_i D + \rho_i^2 F)}{\partial \rho_i} = -D + 2\rho_i F \tag{s-11}$$

The gradient of the marginal log likelihood with respect to $\rho_i$ is :

$$\frac{\partial \log(p(Y_i|L, s_i \sigma_i, \rho_i))}{\partial \rho_i}$$

$$= -\frac{n_{run}\rho_i}{1-\rho_i^2} - \frac{1}{2} Tr[(s_i^2 L^T X^T A_i X L + I)^{-1} \frac{\partial(s_i^2 L^T X^T A_i X L + I)}{\partial \rho_i}]$$

$$+ \frac{1}{2\sigma_i^2}[s_i^2 Y_i^T \frac{\partial A_i}{\partial \rho_i} X L \Lambda_i L^T X^T A_i Y_i + s_i^2 Y_i^T A_i X L \frac{\partial \Lambda_i}{\partial \rho_i} L^T X^T A_i Y_i + s_i^2 Y_i^T A_i X L \Lambda_i L^T X^T \frac{\partial A_i}{\partial \rho_i} Y_i]$$

$$- \frac{1}{2\sigma_i^2} Y_i^T \frac{\partial A_i}{\partial \rho_i} Y_i$$

$$= -\frac{n_{run}\rho_i}{1-\rho_i^2} - \frac{s_i^2}{2} Tr[\Lambda_i L^T X^T \frac{\partial A_i}{\partial \rho_i} X L]$$

$$+ \frac{s_i^2}{\sigma_i^2} Y_i^T \frac{\partial A_i}{\partial \rho_i} X L \Lambda_i L^T X^T A_i Y_i$$

$$- \frac{s_i^4}{2\sigma_i^2} Y_i^T A_i X L \Lambda_i L^T X^T \frac{\partial A_i}{\partial \rho_i} X L \Lambda_i L^T X^T A_i Y_i - \frac{1}{2\sigma_i^2} Y_i^T \frac{\partial A_i}{\partial \rho_i} Y_i$$

Therefore,

$$\frac{\partial \log(p(Y_i|L, s_i \sigma_i, \rho_i))}{\partial a_i}$$

$$= \frac{\partial \log(p(Y_i|L, s_i \sigma_i, \rho_i))}{\partial \rho_i} \frac{\partial \rho_i}{\partial a_i}$$

$$= \frac{2}{\pi(1+a_i^2)} \{ -\frac{n_{run}\rho_i}{1-\rho_i^2} - \frac{s_i^2}{2} Tr[\Lambda_i L^T X^T \frac{\partial A_i}{\partial \rho_i} X L] \tag{s-12}$$

$$+ \frac{s_i^2}{\sigma_i^2} Y_i^T \frac{\partial A_i}{\partial \rho_i} X L \Lambda_i L^T X^T A_i Y_i$$

$$- \frac{s_i^4}{2\sigma_i^2} Y_i^T A_i X L \Lambda_i L^T X^T \frac{\partial A_i}{\partial \rho_i} X L \Lambda_i L^T X^T A_i Y_i - \frac{1}{2\sigma_i^2} Y_i^T \frac{\partial A_i}{\partial \rho_i} Y_i \}$$

### S.3.4 Gradient with respect to L

We start with the gradient with respect to the (j, k) element $L_{jk}$ of $L$ (j ≤ k).

Note that $\frac{\partial L}{L_{jk}} = \delta_j \delta_k^T$

$$\frac{\partial \log\left(p\left(Y_i | L, s_i \sigma_i, \rho_i\right)\right)}{\partial L_{jk}}$$

$$= -\frac{1}{2} Tr\left[\Lambda_i \frac{\partial \Lambda_i^{-1}}{\partial L_{jk}}\right] + \frac{s_i^2}{2\sigma_i^2} \frac{\partial\left(Y_i^T A_i X L \Lambda_i L^T X^T A_i Y_i\right)}{\partial L_{jk}}$$

$$= -\frac{1}{2} Tr\left[\Lambda_i \frac{\partial\left(s_i^2 L^T X^T A_i X L + I\right)}{\partial L_{jk}}\right] + \frac{s_i^2}{2\sigma_i^2} \frac{\partial\left(Y_i^T A_i X L \left(s_i^2 L^T X^T A_i X L + I\right)^{-1} L^T X^T A_i Y_i\right)}{\partial L_{jk}}$$

$$= -\frac{s_i^2}{2} Tr\left[\Lambda_i\left(\delta_k \delta_j^T X^T A_i X L + L^T X^T A_i X \delta_j \delta_k^T\right)\right]$$

$$+ \frac{s_i^2}{2\sigma_i^2} Y_i^T A_i X \delta_j \delta_k^T \left(s_i^2 L^T X^T A_i X L + I\right)^{-1} X^T A_i Y_i$$

$$- \frac{s_i^2}{2\sigma_i^2} Y_i^T A_i X L \left(s_i^2 L^T X^T A_i X L + I\right)^{-1} \frac{\partial\left(s_i^2 L^T X^T A_i X L + I\right)}{\partial L_{jk}} \left(s_i^2 L^T X^T A_i X L + I\right)^{-1} L^T X^T A_i Y_i$$

$$+ \frac{s_i^2}{2\sigma_i^2} Y_i^T A_i X L \left(s_i^2 L^T X^T A_i X L + I\right)^{-1} \delta_k \delta_j^T X^T A_i Y_i$$

$$= -\frac{s_i^2}{2} Tr\left[\delta_j^T X^T A_i X L \Lambda_i \delta_k\right] - \frac{s_i^2}{2} Tr\left[\delta_k^T \Lambda_i L^T X^T A_i X \delta_j\right]$$

$$+ \frac{s_i^2}{2\sigma_i^2} Tr\left[Y_i^T A_i X \delta_j \delta_k^T \Lambda_i X^T A_i Y_i\right]$$

$$- \frac{s_i^4}{2\sigma_i^2} Tr\left[Y_i^T A_i X L \Lambda_i \left(\delta_k \delta_j^T X^T A_i X L + L^T X^T A_i X \delta_j \delta_k^T\right) \Lambda_i L^T X^T A_i Y_i\right]$$

$$+ \frac{s_i^2}{2\sigma_i^2} Tr\left[Y_i^T A_i X L \Lambda_i \delta_k \delta_j^T X^T A_i Y_i\right]$$

$$= -\frac{s_i^2}{2}\{X^T A_i X L \Lambda_i\}_{jk} - \frac{s_i^2}{2}\{\Lambda_i L^T X^T A_i X\}_{kj}$$

$$+ \frac{s_i^2}{2\sigma_i^2} Tr[\delta_k^T \Lambda_i X^T A_i Y_i Y_i^T A_i X \delta_j] + \frac{s_i^2}{2\sigma_i^2} Tr[\delta_j^T X^T A_i Y_i Y_i^T A_i X L \Lambda_i \delta_k]$$

$$- \frac{s_i^4}{2\sigma_i^2} Tr[\delta_j^T X^T A_i X L \Lambda_i L^T X^T A_i Y_i Y_i^T A_i X L \Lambda_i \delta_k] - \frac{s_i^4}{2\sigma_i^2} Tr[\delta_k^T \Lambda_i L^T X^T A_i Y_i Y_i^T A_i X L \Lambda_i L^T X^T A_i X \delta_j]$$

$$= -\frac{s_i^2}{2}\{X^T A_i X L \Lambda_i\}_{jk} - \frac{s_i^2}{2}\{\Lambda_i L^T X^T A_i X\}_{kj}$$

$$+ \frac{s_i^2}{2\sigma_i^2}\{\Lambda_i X^T A_i Y Y_i^T A_i X\}_{kj} + \frac{s_i^2}{2\sigma_i^2}\{X^T A_i Y_i Y_i^T A_i X L \Lambda_i\}_{jk}$$

$$- \frac{s_i^4}{2\sigma_i^2}\{X^T A_i X L \Lambda_i L^T X^T A_i Y_i Y_i^T A_i X L \Lambda_i\}_{jk} - \frac{s_i^4}{2\sigma_i^2}\{\Lambda_i L^T X^T A_i Y_i Y_i^T A_i X L \Lambda_i L^T X^T A_i X\}_{kj}$$

$$= -s_i^2\{X^T A_i X L \Lambda_i\}_{jk} + \frac{s_i^2}{\sigma_i^2}\{X^T A_i Y_i Y_i^T A_i X L \Lambda_i\}_{jk}$$

$$- \frac{s_i^4}{\sigma_i^2}\{X^T A_i X L \Lambda_i L^T X^T A_i Y_i Y_i^T A_i X L \Lambda_i\}_{jk}$$

<div align="right">(s-13)</div>

In the above derivation, we used the properties of $Tr[AB]=Tr[BA]$ and that a number is the trace of itself considered as a one-element matrix. $\{M\}_{jk}$ means the (j,k) element of matrix $M$.

Notice that the derivative with respect to the (j,k) element $L$ is equal to the (j,k) element of the matrix in the last equality. Therefore, we can see that the derivative with respect to $L$ in the matrix form is:

$$\frac{\partial \log\left(p\left(Y_i|L, s_i \sigma_i, \rho_i\right)\right)}{\partial L}$$

$$= -s_i^2 X^T A_i X L \Lambda_i + \frac{s_i^2}{\sigma_i^2} X^T A_i Y_i Y_i^T A_i X L \Lambda_i \qquad \textbf{(s-14)} ,$$

$$- \frac{s_i^4}{\sigma_i^2} X^T A_i X L \Lambda_i L^T X^T A_i Y_i Y_i^T A_i X L \Lambda_i$$

for the lower-triangular part.

## S.4 Gaussian Process prior on log(s)

The kernel of the Gaussain Process (GP) we use is the product of a squared exponential kernel defined on the spatial coordinates and one defined on the mean intensity of each voxel.

We denote $W_{space}$ as the matrix of squared spatial distance between voxels, and $W_{inten}$ as the matrix of squared intensity difference between voxels.

Then, in our GP prior, the random vector $\log(s) \in \mathbb{R}^{n_s}$ follows multivariate Gaussian distribution of

$$\log(s) \sim N\left(0, \tau^2 K\right) \qquad \textbf{(s-15)},$$

where

$$K = \exp\left[-\frac{1}{2}\left(\frac{W_{space}}{l_{space}^2} + \frac{W_{inten}}{l_{inten}^2}\right)\right] + \eta I \qquad \textbf{(s-16)}$$

$\eta$ is a small number added to the diagonal of the covariance kernel, in order to guarantee matrix $K$ is invertible. In our implementation, we set it to 0.0001

$l_{\text{space}}$ and $l_{\text{inten}}$ are length scales of the squared exponential kernel for spatial distance and intensity difference, respectively. $\tau^2$ is the variance of the GP.

This prior on log(s) adds additional term to the posterior of the parameters.

$$p(L,\sigma,\rho,s,\tau,l_{space},l_{inten}|Y) \propto p(Y|L,\sigma,\rho,s)P(s|\tau,l_{space},l_{inten})P(\tau)P(l_{space})P(l_{inten})$$

We assume uniform prior for other parameters: $L$, $\sigma$ and $\rho$, because empirically $\sigma$ and $\rho$ can be recovered well, and we do not have a good principle to regularize $L$.

To fit the model, we optimize the joint probability of both the parameters $L$, $\sigma$, $\rho$ and $s$, and the hyper-parameters $\tau$, $l_{\text{space}}$ and $l_{\text{inten}}$.

We parametrize $l_{\text{space}}$ and $l_{\text{inten}}$ with $\log(l_{\text{space}}^2)$ and $\log(l_{\text{inten}}^2)$ to keep all parameters unconstrained. In practice, the optimization of parameters can be instable without regularizing $\tau$, $l_{\text{space}}$ and $l_{\text{inten}}$. Therefore, we introduced weakly informative half-Cauchy prior on these hyper-parameters:

$$p(\tau)=\frac{2\gamma_\tau}{\pi}\frac{1}{\tau^2+\gamma_\tau^2} \quad , \quad p(l_{space})=\frac{2\gamma_{l_{space}}}{\pi}\frac{1}{l_{space}^2+\gamma_{l_{space}}^2} \quad , \quad p(l_{inten})=\frac{2\gamma_{l_{inten}}}{\pi}\frac{1}{l_{inten}^2+\gamma_{l_{inten}}^2}$$

The scale parameters $\gamma$ are set with reasonably large values which the user believe covers the plausible range of $\tau$, $l_{\text{space}}$ and $l_{\text{inten}}$. Since the GP is defined for log(s), $\gamma_\tau = 5$ is a reasonable scale parameter ( $e^5 \sim 150$ ). We set $\gamma_{l_{spacec}}$ and $\gamma_{l_{inten}}$ as half of the maximal distance in space or intensity between all voxels in the ROI.

Therefore, the following term is added to the log of joint probability of all parameters (neglecting constant terms):

$$-\frac{n_S}{2}\log(2\pi\tau^2)-\frac{1}{2}\log|K|-\frac{1}{2\tau^2}\log(s)^T K^{-1}\log(s)-\log(\tau^2+\gamma_\tau^2)-\log(l_{space}^2+\gamma_{l_{space}}^2)-\log(l_{inten}^2+\gamma_{l_{inten}}^2)$$

**(s-17)**.

$-\frac{1}{2\tau^2}K^{-1}\log(s)$ is added to the gradient with respect to $\log(s^2)$.

$$\frac{\partial p(L,s_i\sigma_i,\rho_i,\tau,l_{space},l_{inten}|Y)}{\tau^2}$$
$$=-\frac{n_s}{2\tau^2}+\frac{\log(s)^T K^{-1}\log(s)}{2\tau^4}-\frac{1}{\tau^2+\gamma_\tau^2}$$

The positive solution of the above term being equal to zero provides the maximum a posterior estimate of $\tau^2$ given other parameters:

$$\tau_{ML}^2=\frac{\log(s)^T K^{-1}\log(s)-n_s\gamma_\tau^2+\sqrt{n_S^2\gamma_\tau^4-(2n+8)\gamma_\tau^2\log(s)^T K^{-1}\log(s)+[\log(s)^T K^{-1}\log(s)]^2}}{2(n_S+2)}$$

**(s-18)**

The gradient with respect to $\log(l_{\text{space}}^2)$ is:

$$\frac{\partial p(L,s,\sigma,\rho,\tau,l_{space},l_{inten}|Y)}{\log(l_{space}^2)}$$

**(s-19)**

$$=l_{space}^2\{\frac{1}{2\tau^2}\log(s)^T K^{-1}\frac{\partial K}{\partial l_{space}^2}K^{-1}\log(s)-\frac{1}{2}Tr[K^{-1}\frac{\partial K}{\partial l_{space}^2}]-\frac{1}{l_{space}^2+\gamma_{l_{space}}}\}$$

where the (j,k) element of $\dfrac{\partial K}{\partial l^2_{space}}$ is

$$\{\frac{\partial K}{\partial l^2_{space}}\}_{jk}=\frac{1}{2\,l^4_{space}}\{W_{space}\}_{jk}\cdot\{\exp[-\frac{1}{2}(\frac{W_{space}}{l^2_{space}}+\frac{W_{inten}}{l^2_{inten}})]\}_{jk} \tag{s-20}$$

And

$$\frac{\partial\,p(L,s,\sigma,\rho,\tau,l_{space},l_{inten}|Y)}{\log(l^2_{inten})}$$
$$=l^2_{inten}\{\frac{1}{2\,\tau^2}\log(s)^T K^{-1}\frac{\partial K}{\partial l^2_{inten}}K^{-1}\log(s)-\frac{1}{2}Tr[K^{-1}\frac{\partial K}{\partial l^2_{inten}}]-\frac{1}{l^2_{inten}+\gamma_{l_{inten}}}\} \tag{s-21}$$

$$\{\frac{\partial K}{\partial l^2_{inten}}\}_{jk}=\frac{1}{2\,l^4_{inten}}\{W_{inten}\}_{jk}\cdot\{\exp[-\frac{1}{2}(\frac{W_{space}}{l^2_{space}}+\frac{W_{inten}}{l^2_{inten}})]\}_{jk} \tag{s-22}$$

Practically, the smoothness of pseudo-SNR is usually over-estimated, sometimes resulting to almost equal estimation of pseudo-SNR over all voxels. Therefore, we can also consider using an inverse Gamma distribution as a prior for $\tau^2$: $\quad \tau^2\sim inv\text{-}Gamma(\alpha_{\tau^2},\beta_{\tau^2})$ Then, (s-17) becomes

$$-\frac{n_S}{2}\log(2\pi\tau^2)-\frac{1}{2}\log|K|-\frac{1}{2\,\tau^2}\log(s)^T K^{-1}\log(s)-(\alpha_{\tau^2}+1)\log(\tau^2)-\frac{\beta_{\tau^2}}{\tau^2}-\log(l^2_{space}+\gamma^2_{l_{space}})-\log(l^2_{inten}+\gamma^2_{l_{inten}}) \tag{s-23}$$

$$\frac{\partial\,p(L,s_i\,\sigma_i,\rho_i,\tau,l_{space},l_{inten}|Y)}{\tau^2}$$
$$=-\frac{n_s}{2\,\tau^2}+\frac{\log(s)^T K^{-1}\log(s)}{2\,\tau^4}-\frac{(\alpha_{\tau^2}+2)}{\tau^2}+\frac{\beta_{\tau^2}}{\tau^4} \tag{s-24}$$

So

$$\tau^2_{MAP}=\frac{2\beta_{\tau^2}+\log(s)^T K^{-1}\log(s)}{2\,\alpha_{\tau^2}+2+n_s} \tag{s-25}$$

## S.5 Bias introduced by applying standard RSA on raw fMRI data

Instead of building design matrix that reflects hypothetical hemodynamic response, researchers sometimes take the raw fMRI signal (sometimes after pre-processing such as despiking and detrending) approximately 6 seconds after an event as the neural response "pattern" to that event. As mentioned in the main text at the end of **Section 2**, such approach also suffers from bias. The expected covariance matrix between such patterns is

$$(X^T X)^{-1}X^T X_{true}U X^T_{true}X(X^T X)^{-1}+(X^T X)^{-1}X^T \Sigma_\epsilon X(X^T X)^{-1} \tag{s-26}.$$

Where $X_{true}$ is the design matrix reflecting the true HRF in fMRI, and $X$ is a design matrix which has a single-pulse 6 sec after each event. Such a design matrix with single-pulses is an implicit assumption when treating the raw fMRI data as neural pattern. It is easy to observe that averaging patterns of the same condition following this approach is equivalent as calculating $\hat{\beta}$ with equation (2) but assuming single-pulse HRF.

The first term in (s-26) is introduced by implicitly assuming an unrealistic HRF. Because of this assumption, the "patterns" obtained are in fact mixtures of the true neural patterns of adjacent neural events. Therefore, it alters the correlation between estimated patterns of temporally proximate neural events. To illustrate this bias, we simulate data which is generated by weighting the true design matrix reflecting realistic HRF with $\beta$'s following the covariance structure in Fig 2A, but without adding any noise. If there were no noise, standard RSA on the estimates of $\beta$'s would not introduce bias. However, with this approach on raw simulated data even without noise, we observe structured bias unrelated to the true covariance matrix shown in **Fig S-1**, which is due to the first term in (s-26).

Figure S-1. A bias structure is obtained by applying standard RSA on the raw patterns of simulated data without noise added.

The second term in (s-26) is introduced because of the autocorrelation structure of fMRI noise, together with the event timing. To illustrate this bias, we applied the raw pattern approach to preprocessed resting state fMRI data of 30 participants in the Human Connectome Project (HCP), pretending the participants took part in the task in **Fig 1A**. These data have no relation with the task, so there is no chance of measuring any real neural activity related to the task. However, we observe a similarity structure with many high correlations, as shown in **Fig S-2**. This clearly illustrates the bias introduced by treating raw fMRI data as neural patterns of events.

Figure S-2. A bias structure is obtained by applying standard RSA on the raw patterns of an unrelated resting state fMRI dataset.

Data used for the illustration in **S.5** were provided by the MGH-USC Human Connectome Project (HCP; Principal Investigators: Bruce Rosen, M.D., Ph.D., Arthur W. Toga, Ph.D., Van J. Weeden, MD). HCP funding was provided by the National Institute of Dental and Craniofacial Research (NIDCR), the

National Institute of Mental Health (NIMH), and the National Institute of Neurological Disorders and Stroke (NINDS). HCP data are disseminated by the Laboratory of Neuro Imaging at the University of California, Los Angeles.