[Reviews · NeurIPS 2016]

Reviewer 1

Summary

This manuscript address a very important topic of bias in representational similarity measures derived from fmri signals. The authors demonstrate analytically and through numerical simulations that bias arises because of noise in fmri signals and because fmri responses to subsequent stimuli overlap and are correlated in time. The proposed unbiased solution involves a maximum likelihood estimation of the covariance matrix from data and then using this matrix to compute the representational similarity. The method also returns an estimate of variability at different voxels; less relaible voxels contribute less to the total log-likelihood.

Qualitative Assessment

This is fantastic paper in all respects: it addresses an important point, includes analytics, numerical simulations on model data and analysis of real data. The paper is also clearly written.

Confidence in this Review

3-Expert (read the paper in detail, know the area, quite certain of my opinion)


Reviewer 2

Summary

The paper investigates the issue of bias introduced in the traditional estimation of similarity matrix of neural activity patterns. The paper adheres to a simple linear formulation for the production of fMRI time series, and shows that in such a case, a solution obtained by minimizing the mean-square is biased if we were to assume a multivariate Gaussian distribution for voxel responses to different conditions. The authors then present a Bayesian inference formulations to directly estimate the similarity matrix without resorting to the intermediate (biased) point estimates.

Qualitative Assessment

The use of a Bayesian approach as opposed to a point estimate is interesting, and not surprisingly does better at decoupling the different uncertainties. Therefore, even though the contribution is novel in the problem domain, the idea and tools used are well understood. In my opinion, this is a good application of a Bayesian modeling framework, but it does not lead to any deep insights or development of better tools. Following are some of the points I would like to make: 1. The presentation of the ideas in the text is quite good, and easy to follow. However, the figures are difficult to understand. I did not quite understand the task as shown by Fig. 1A, and the counterpart in case of noise in 1C. The heat maps of the similarity matrices in Fig 1B,D are also not very informative. Sure the diagonal elements have high values in both, but how do we compare others and reach the conclusion that they are similar? Same concern holds for Fig. 2A, B and Fig. 3A,B. Specifically, again 3B is claimed to "correspond nicely to the behaviorally-reported structure" with no quantitative reasoning especially since even the heat maps don’t look much alike. Fig. 2C is small, difficult to read and has different scales on the 3 y-axes making it visually confusing and less informative. 2. Some of the assumptions in the generative model are made without much justification. For example, assuming that the voxel responses are normally distributed. In summary, for most part the development of the paper is good, and even though there is some novelty in using it for the problem domain, there is no novelty to the methodology (Bayesian inference in a simple graphical model), and the results on the real dataset are unconvincing.

Confidence in this Review

2-Confident (read it all; understood it all reasonably well)


Reviewer 3

Summary

Authors show that RSA suffers from bias that is introduced by the experimental design and the structured noise in neutral estimates. The Bayesian RSA method proposed by the authors estimates the noise covariance from data minimizing the impact of this structured noise.

Qualitative Assessment

The topic is interesting and significant. It is important to point at confounding parameters. Motivation, method description and the toy data example are clearly described. I also agree wit the training of the authors. The real world example, however, would benefit from a more detailed discussion. I understand that the results are "nice" (caption figure 3(b). What does this mean? Did authors gain new knowledge or are the results better fitting the authors expectations?

Confidence in this Review

2-Confident (read it all; understood it all reasonably well)


Reviewer 4

Summary

The paper reveals that when representational similarity analysis (RSA) is conducted using regression weights estimates, the resulting similarity matrix can suffer from spurious bias structure. The authors introduce a novel method for calculating RSA, called “Bayesian RSA”, which estimates the shared covariance structure of activity patterns directly from the data, instead of relying on the regression weights estimates. Simulated data results from the autoregressive first-order noise model suggest that Bayesian RSA can recover shared covariance structure of activity patterns with less bias than vanilla RSA under this model. Experiments on one real subject suggest that the similarity matrix estimated by Bayesian RSA from fMRI data is somewhat similar to the “ground truth” subjectively judged similarity matrix.

Qualitative Assessment

The paper explains well how computing RSA using estimates of regression weights can result in a biased similarity matrix. However, in many cases in neuroscience, the RSA is computed directly on the patterns of activity, and not the estimates of regression weights beta. This diminishes the relevance of this paper to the neuroscience field. The authors very briefly address this alternate way of computing RSA in lines 123-128. It is unclear how this alternative RSA computation is biased if it does not depend on a proxy for beta estimates, and needs to be addressed further. One way to empirically address this concern would be to add a comparison to an RSA estimated directly from the activity patterns in Figure 2. The synthetic data was simulated using the same noise model (AR(1)) that the Bayesian RSA assumes is true (see line 140 for the assumption). Thus, the results in Figure 2B showing that Bayesian RSA can recover the shared covariance structure with less bias than vanilla RSA under the true assumed noise model are positive, but not surprising. A much more convincing experiment would be a comparison under different types of noise models that violate the Bayesian RSA assumption, especially since an AR(1) noise model is not guaranteed to be true in practice. The results from the real neuroscience data (fMRI recordings) for the Bayesian RSA should also be compared to an RSA computed directly with the pattern activities. In addition, the data set that the authors used contained more than one participant’s fMRI recordings, so the fact that the authors chose to display the Bayesian RSA results for only one participant (as opposed to several participants or an average over participants) is disappointing. Figure 3C would make a stronger case had it been compared with estimates of regression weights from state-of-the-art methods (e.g. kernel regression). Overall, the paper would have benefited from an objective measure of similarity between different RSA matrices (e.g. rank correlation) There were also some typos: line 134, “not relying analysis on this”; line 219, “an alysis”

Confidence in this Review

3-Expert (read the paper in detail, know the area, quite certain of my opinion)


Reviewer 5

Summary

The manuscript proposes a new approach to Representational Similarity Analysis (RSA) that is based on learning parameters of a model that accounts for correlated noise between pixels and through time. The proposed methods empirically improves the covariance estimate of \beta by greatly reducing the bias that is used in RSA.

Qualitative Assessment

I have one major complaint, which is that the authors claim that they have "unbiased" estimation, but this is never proven. I do not believe the method is unbiased, however, practically this method seems to greatly reduce the bias. Either the claim of unbiased needs to be proven, or the claimed should be changed that the method empirically reduces the bias. Minor complaints: The parameter \beta is switched around in language between a matrix and a vector. This needs to be cleaned up in the revision. It is not clear that is is "reasonable" to assume that \beta is independent from the noise. Either give justification or change this to simply state that this is an assumption. It should be clarified that \hat\beta itself in section 2 is not biased, and the bias is only on the covariance (which is in turn used for estimating similarity).

Confidence in this Review

2-Confident (read it all; understood it all reasonably well)


Reviewer 6

Summary

The manuscrit presents a technical bias that has impaired accurate estimation of the structure of similiarity between neural representations, i.e. the similarity between neural representations of distinct objects or percepts, from neuroimaging data (especially fMRI data). The manuscript first details mathematically how the bias emerges in the covariance matrix, especially when signal-to-noise ratio is low and the timing of presentations of the different objects are not well balanced. Then authors propose a new method for structure inference based on maximum likelihood estimation from a generative model of neural data with various parameters (covariance structure, SNR in all voxels, variance and autoregressive coefficient of noise). This method naturally gets rid of the estimation bias, as is shown with synthetic data. It is finally applied to a fMRI dataset showing similarity between a behaviourally derived and a neurally derived similarity structure of representations of animals in human subjects.

Qualitative Assessment

The manuscript is very clearly written, the method is impressive from the technical point of view and it seems to address very appropriately a real issue for analysis of neuroimaging data. I expect that it can be of great use to many neuroscientists by improving the estimation of similarity structure from neural activity. Moreover the perspective of having improved measure of betas in fMRI (notably taking into account the bias in covariance matrix and the smooth prior for SNR) is extremely promising. This being said, here is a series of additional comments: -The authors argue that alternate methods are not well grounded, but it would be nice to have the illustration in Figure 2 that the new method actually performs better at recovering the covariance structure in synthetic data. - I have a concern regarding the spatial independence of noise. If noise is not spatially independent (and it generally isn't, as acknowledged in the discussion), this will create some consistent overestimation the correlational structure. Will this affect more the estimation than the original bias that the method seeks to get rid off ? In other terms, is the cure worse than the disease ? - It would be great if the code could be made public, and perhaps interfaced with one of the popular fMRI analyses softwares/toolboxes such as SPM. - maybe in figure 3 correlation between behaviourally and neural obtained matrices could be used as a measure of the correspondance between the two. *Writing/typos: - L35-36 “ what stimuli have very different representations” → rephrase - L41 “demonstrations” -L42 “highly similar representational structure” unclear - L59-60 maybe specify that when Y is white noise, there is still structure in design matrix X (which is where the 'representational' structure comes from). It cost me quite a lot to understand that point. - L94 “is”->”are” L214 “advantage not available” → either “advantage over” or “feature not available” L230 “intensities” L250 “experiment design” [EDIT AFTER REBUTTAL] all my concerns were convincingly addressed, except the one about testing the bayesian RSA against alternative methods described l268-287 on synthetic data.

Confidence in this Review

2-Confident (read it all; understood it all reasonably well)